# Evolving Practices in Low-Risk Papillary Thyroid Cancer: Impact of the 2015 ATA Guidelines

**DOI:** 10.3390/curroncol33010026

**Published:** 2026-01-02

**Authors:** Benard Gjeloshi, Leonardo Rossi, Carlo Enrico Ambrosini, Chiara Becucci, Piermarco Papini, Andrea De Palma, Luigi De Napoli, Marco Puccini, Gabriele Materazzi

**Affiliations:** Department of Surgical, Medical and Molecular Pathology and Critical Area, University of Pisa, 56124 Pisa, Italy; leonardo.rossi1@unipi.it (L.R.); carloeambrosini@gmail.com (C.E.A.); chiara.becucci@phd.unipi.it (C.B.); piermarco.papini@phd.unipi.it (P.P.); a.depalma@studenti.unipi.it (A.D.P.); l.denapoli@hotmail.it (L.D.N.); marco.puccini@unipi.it (M.P.); gabriele.materazzi@unipi.it (G.M.)

**Keywords:** thyroid, papillary carcinoma, trend, low risk, thyroidectomy, lobectomy, ATA guidelines

## Abstract

This study examined how surgical management of low-risk papillary thyroid carcinoma (PTC) changed after the 2015 American Thyroid Association guidelines recommended more conservative approaches. Researchers retrospectively analyzed 1644 patients treated between 2014 and 2023. Over time, thyroid lobectomy (TL) increasingly replaced total thyroidectomy (TT), rising from 0% in 2014 to nearly 60% in 2023, especially for microcarcinomas and tumors up to 2 cm. TT remained common for lesions larger than 2 cm. The need for completion thyroidectomy after TL dropped sharply. Compared with TL, TT showed higher rates of overall complications, including transient and permanent hypoparathyroidism, and required longer operative time and hospitalization. As TL became more widely adopted, postoperative hypoparathyroidism decreased. Overall, the findings indicate a gradual but clear shift toward less extensive surgery for low-risk PTC, supporting the safety and clinical benefits of the guideline-recommended conservative approach.

## 1. Introduction

Papillary thyroid carcinoma (PTC) is the most common type of differentiated thyroid cancer [1]. Over the past several decades, the incidence of PTC has risen substantially worldwide, largely attributed to increased detection of small, indolent tumors through widespread use of imaging and fine-needle aspiration biopsy [2]. Despite this rise, disease-specific mortality has remained very low. This makes PTC amenable to less invasive treatment modalities with excellent oncologic outcomes. Some PTCs are considered “low risk”, based on size, histological characteristics, and nodal status [3].

Traditionally, total thyroidectomy (TT) has been the standard surgical treatment for differentiated thyroid cancer, including low-risk PTC, based on the rationale that complete thyroid removal facilitates radioactive iodine (RAI) therapy and serum thyroglobulin surveillance. However, TT carries a higher risk of complications such as hypoparathyroidism and recurrent laryngeal nerve injury, particularly when compared with thyroid lobectomy (TL). In recent years, accumulating evidence has demonstrated that TL provides equivalent oncological outcomes for properly selected patients with low-risk PTC, while offering the advantage of reduced perioperative morbidity [4,5].

In recognition of these findings, the 2015 American Thyroid Association (ATA) guidelines endorsed TL as an acceptable initial surgical option for intrathyroidal PTC measuring 1–4 cm, without high-risk features [6]. Although this recommendation represented a significant shift toward more conservative management, its adoption has varied widely among institutions and geographic regions [3,7]. Factors influencing surgeon preference include concerns about the need for completion thyroidectomy, perceived oncologic adequacy, and familiarity with traditional treatment paradigms [8,9,10]. Accurate risk stratification for cancer recurrence is critical to shared decision-making [11,12]. Information regarding available treatment options for clinically low-risk PTC as well as the risk of thyroidectomy and postoperative quality of life should be thoroughly discussed during initial counseling [13].

Understanding real-world patterns of care is essential for assessing guideline uptake and its impact on patient outcomes. However, evidence regarding how quickly and to what extent the 2015 ATA recommendations have been implemented—particularly in high-volume centers—is still limited. Moreover, contemporary data examining the effect of changing surgical practices on postoperative complications remain sparse [14,15].

The present study aims to evaluate temporal trends in the initial surgical management of low-risk PTC in a high-volume tertiary center over a ten-year period and to assess how the adoption of ATA guideline recommendations has influenced postoperative outcomes. By analyzing a large cohort of consecutive patients, we seek to clarify the evolution of clinical practice and provide insights into the real-world transition toward a less aggressive surgical approach.

## 2. Materials and Methods

This is a single-center, retrospective, observational study. All patients who underwent surgery for thyroid disease at the Endocrine Surgery Unit of Pisa University Hospital between January 2014 and December 2023 were initially considered.

This study included patients with localized, low-risk papillary thyroid carcinoma or microcarcinoma.

Inclusion criteria were preoperative diagnosis of differentiated thyroid carcinoma (TIR4 or TIR5 according to the Italian Consensus Statement for the Classification and Reporting of Thyroid Cytology (SIAPEC-IAP classification) [16], unifocal, intrathyroidal nodule; tumor size < 4 cm, age ≥ 18 years, and absence of clinical or radiological evidence of vascular or macroscopic extrathyroidal invasion, cervical lymph node or distant metastasis.

Only patients who underwent initial surgery for PTC and meeting the 2015 ATA criteria for potential eligibility for lobectomy were included.

Patients with a history of neck radiation, a family history of thyroid carcinoma, or previous thyroid surgery were excluded from this study.

The preoperative diagnostic workup included clinical evaluation, neck ultrasound, thyroid hormone assays, and fine-needle aspiration cytology (FNAC) results which were classified according to SIAPEC-IAP classification. Further imaging procedures (e.g., CT scan) were performed when deemed necessary based on clinical judgment.

The algorithm used for patient selection is illustrated in Figure 1.

Patients were categorized into two groups according to the first course-treatment:

**Group A:** 362 patients who underwent TL.

**Group B:** 1282 patients who underwent TT as the initial approach.

Although the inclusion criteria were the same for both groups, the choice between TL and TT was not random but based on preoperative clinical evaluation and reflected routine clinical practice. Surgical strategy was determined by endocrine surgeons considering tumor characteristics, imaging findings, and patient-related factors.

All procedures were performed by experienced high-volume endocrine surgeons, each performing more than 100 thyroid operations per year, in accordance with established definitions of surgical expertise [17,18].

We evaluated treatment trends for localized PTC during the study period, stratified by tumor size (<1 cm, 1 to <2 cm, 2 to <4 cm), age at diagnosis (<55 or ≥55 years), and sex. In patients who underwent TL as the initial surgical approach, we further assessed the rate of completion thyroidectomy after definitive histopathological examination.

For each patient, the following variables were collected: sex, age, nodule size, estimated thyroid volume, FNAC category, type of surgical procedure, operative time (defined as time from skin incision to skin closure), postoperative length of hospital stay, postoperative complications (transient and/or permanent hypocalcemia, transient and/or permanent paralysis of the recurrent laryngeal nerve (RLN), cervical bleeding).

Hypocalcemia was defined as a serum calcium level < 8.0 mg/dL or the need for ongoing replacement therapy with calcium carbonate and/or calcitriol. Unilateral RLN palsy was diagnosed in case of documented impairment of cord mobility observed during fiberoptic laryngoscopy. A 6-month cutoff was used to differentiate transient from permanent postoperative hypocalcemia and RLN palsy. Postoperative bleeding was defined as any hemorrhage requiring treatment, either through surgical revision or conservative management of cervical hematoma.

This study was conducted in accordance with the principles of the Declaration of Helsinki, and all patients provided written informed consent. This study has been approved by the local ethical committee, Comitato Etico Area Vasta Nord-Ovest (CEAVNO) (CET40/2025).

### Statistical Analysis

The Shapiro–Wilk test was used to test the normality of the numerical variables. Continuous quantitative data were expressed as mean ± standard deviation (SD) or medians and interquartile ranges (IQR) and compared using Student’s *t* test or Mann–Whitney U test, as appropriate. Categorical qualitative data were expressed as numbers and percentages and compared using the χ^2^ test (or Fisher’s exact test, as applicable). These analyses were carried out with SPSS v.26 (IBM Corp., Armonk, NY, USA).

To statistically assess the significance of observed shifts in treatment proportion trends, we employed Joinpoint regression (Joinpoint Regression Program, version 5.0.2) to estimate annual percentage changes and identify inflection points. The statistical significance for all performed analyses was set at *p* < 0.05.

## 3. Results

During the study period, 27,269 patients underwent surgery for thyroid disease.

Overall, 1644 (6%) patients were included in this study. Considering the initial surgical approach, 326 patients (21.8%) underwent TL, while 1282 (78.2%) received TT.

Based on nodule diameter, 443 (26.9%) were diagnosed with papillary thyroid microcarcinoma (tumor size < 1 cm) and 1201 (73.1%) had localized low-risk papillary carcinoma (tumor size 1–4 cm).

Group A (TL) and Group B (TT) did not differ significantly in terms of mean age at diagnosis (41.1 ± 13.4 vs. 40.9 ± 13.1 years; *p* = 0.94), sex distribution (female: 73% vs. 74.6%; *p* = 0.32), body mass index (BMI) (24.1 ± 4.1 kg/m^2^ vs. 24.8 ± 4.6 kg/m^2^; *p* = 0.11) or preoperative FNAC findings (*p* = 0.71).

In contrast, significant differences were observed between the two groups in estimated thyroid volume (11.9 ± 2.3 mL vs. 14.3 ± 4.9 mL; *p* < 0.001) and nodule size (10.8 ± 4.3 mm vs. 15.2 ± 7.3 mm; *p* < 0.001) (Table 1).

Patients in Group B were less likely to undergo a minimally invasive or remote-access approach than those in Group A (10.1% vs. 27.9%; *p* < 0.001).

Postoperative outcomes are summarized in Table 2. Higher complication rates were observed in Group B compared with Group A (12.4% vs. 3.0%; *p* < 0.001), including higher rates of transient hypoparathyroidism (8.9% vs. 0%; *p* < 0.001) and permanent hypoparathyroidism (1.8% vs. 0%; *p* = 0.03). TT patients experienced longer operative time (50.9 ± 15.9 vs. 44.4 ± 20.1 min; *p* < 0.001) and longer postoperative length of stay (1.29 ± 0.6 vs. 1.14 ± 0.4 days; *p* < 0.001). No statistically significant differences were observed between the two groups regarding transient unilateral RLN palsy (2.7% vs. 1.1%; *p* = 0.08) or cervical hematoma (0.9% vs. 1.9%; *p* = 0.11). Notably, four cases of contralateral RLN palsy occurred among patients who underwent TT.

Histopathological examination revealed that multifocality was more frequently observed in Group B (34.1% vs. 24.1%; *p* < 0.001), as was chronic lymphocytic thyroiditis (41.3% vs. 31.2%; *p* < 0.001).

The distribution of TL according to tumor size was evaluated. TL utilization decreased significantly with increasing tumor size (35.8% for T1a, 21.1% for T1b, and 5.5% for T2; *p* < 0.001) (Figure 2), although TT remained the predominant approach in all categories.

When stratified by age, 84.4% of patients were <55 years old and 15.6% were >55 years old. The choice of initial surgical approach did not differ significantly between the two groups (TL 21.6% vs. 23.8%, respectively; *p* = 0.44).

Temporal trends over the study period are presented in Figure 3 and Table 3. Prior to the publication of the ATA 2015 guidelines, nearly all patients were treated with TT. After 2016, the proportion of patients treated with TL progressively increased. A rapid increase was observed during the initial years following the guideline release (23.3% in 2018 vs. 0% in 2014; *p* < 0.001), followed by a plateau phase and a subsequent rise after 2021. In 2023, TL was performed in 60% of cases, representing a significant increase compared with 2022 (*p* < 0.001). This trend was consistent across sex and age (Figure 4 and Figure 5). Among patients with PTC measuring 0–2 cm, the increase in TL mirrored the overall pattern, with major changes occurring shortly after guideline publication and again after 2020. For tumors measuring 2–4 cm, changes were more limited in earlier years, but the TL rate increased to 27% in 2023 (Figure 6).

Among patients who initially underwent TL, the overall completion thyroidectomy rate was 13.8%. Completion rates were similar for T1a and T1b tumors (13% and 13.8%, respectively), and higher for T2 tumors (22.2%), although this difference was not statistically significant (*p* = 0.34) (Table 4). The completion rate was 32% during the early years following the publication of the ATA guidelines but progressively declined after 2020, reaching 2.9% in 2023.

The overall complication rate of the population decreased progressively throughout the study period, from 13.5% to 5.8%, with postoperative hypoparathyroidism following a similar downward trend (Figure 7). Other complications, including vocal cord dysfunction and postoperative bleeding, showed only minor fluctuations over time.

## 4. Discussion

The optimal initial surgical approach for PTC remains debated. Determining the extent of surgery in clinical practice is complex and influenced by several factors, including recurrence risk, the need for adjuvant RAI therapy, procedure morbidity, and patient preference. Tumor-specific characteristics also play a crucial role in surgical decision-making [19].

TT offers the advantage of facilitating postoperative RAI, reliable thyroglobulin monitoring, and addressing synchronous tumors. On the other hand, TL has lower perioperative morbidity, shorter operative time, and reduced hospital stays and potentially requires no levothyroxine supplementation. Despite these benefits and the updated guidelines favoring TL in low-risk PTCs, surgeons hesitate to adopt these recommendations, leading to significant variations in management across centers [8,9,10,20].

Based on estimates of thyroid surgeries in Italy, this study covers approximately 6.8% of all thyroid operations performed during the study period [21]. Our findings revealed that only 22% of the patients underwent TL. As expected, patients undergoing TL experienced lower rates of hypocalcemia, shorter operative time, and shorter hospital stay.

However, TT remains a valid option, particularly when postoperative RAI therapy is anticipated, despite its higher morbidity. When assessing surgical strategies, both survival and recurrence risk must be considered [5,22]. A major concern following TL is the potential presence of pathological high-risk features (HRF) that may upgrade the risk category from low to intermediate. These include aggressive subtype, vascular invasion, extrathyroidal extension, lymph node invasion, or positive margins. In these circumstances, completion thyroidectomy may be needed to facilitate RAI administration [23]. In our study, the completion thyroidectomy rate was 13.8%, consistent with the 5–20% range reported in the literature, reflecting variations in institutional RAI practices [24,25].

As PTC is frequently multifocal, TT may increase the likelihood of identifying synchronous carcinomas. In our cohort, incidental microPTCs were more commonly found in the TT group (*p* < 0.01); however, the clinical relevance of these findings appears limited in appropriately selected patients. Several studies report local recurrence rates of less than 1–4% in experienced centers [26,27]. Moreover, salvage therapy for patients with low-risk disease who experience recurrence after low-intensity therapy has been shown to result in excellent survival rates comparable to those of patients undergoing upfront high-intensity therapy [28].

The role of postoperative RAI therapy continues to be one of the most controversial issues in the management of differentiated thyroid cancer. Evidence regarding the impact of RAI on recurrence and disease-specific survival—particularly among intermediate-risk patients—is inconsistent [29,30,31]. As a result, most guidelines offer broad recommendations for intermediate-risk patients while giving definitive advice only for patients with very low- or high-risk disease [6,32]. In recent years, increasing evidence has supported a more selective use of RAI, influenced by institutional preferences, and evolving clinical practice patterns [33]. This variability has led to differences in the use of TL and completion thyroidectomy across institutions and over time [34].

Our findings highlight the evolving surgical approaches for PTC following the 2015 ATA guidelines. TL rates progressively increased post-guideline implementation, initially plateauing at 20% for three years before surging to 60% by 2023. This trend reflects growing confidence in TL as a safe and effective option for low-risk PTC and aligns with findings from other studies [15,35].

Stratification by tumor size reveals a clear relationship between nodule diameter and surgical choice. As the size of the nodule increased, the proportion of TL decreased (36% for nodules < 1 cm; 21% for nodules 1–2 cm; and 6% for nodules 2–4 cm). The decreasing preference for TL in larger tumors may be attributed to concerns about recurrence risk and the lack of consensus on the classification of these tumors as “low-risk” [36].

The 2009 ATA guidelines [37] recommended TL as an option for selected very low-risk tumors (e.g., unifocal, localized PTC < 1 cm with no aggressive features). Despite this, the rate of TL for these lesions remained very low until 2015, when updated guidelines recommended TL as the preferred treatment for PTC smaller than 1 cm. This led to a significant increase in the use of TL, consistent with global trends [38,39]. After an initial plateau, TL rates rose sharply again, reaching 72% by 2023. A similar pattern emerged for T1b tumors; although only 23% of patients with 1–2 cm PTC underwent TL in 2021, the rate exceeded 50% by 2023, mirroring expanding evidence of TL safety in low-risk disease [40,41]. For T2 tumors, only minor changes were observed until 2022, when TL rates began to increase significantly. Nonetheless, the surgical management of this group remains controversial, and further studies are needed to clarify the best approach.

The impact of the adoption of the 2015 ATA guidelines on completion thyroidectomy is not well-documented in the literature. While Worral et al. reported increases in both TL and completion thyroidectomy following the 2015 ATA guidelines [42] more recent studies have found decreasing completion rates despite broader use of TL [43,44]. In line with these findings, our data demonstrate a marked decrease in completion thyroidectomy rates at our institution, from 36% in 2016 to just 4% in 2023. This likely reflects a more selective use of RAI at our center and improved patient selection for TL.

Age remains one of the most influential prognostic factors in patients with PTC. Some studies suggest that patients ≥ 55 years old may benefit more from TT [45], though similar trends in surgical management have been reported across age groups since 2015 [15,38]. In our study, TL use progressively increased among patients aged ≥ 55 years, reaching 50% in 2023. These demographic trends highlight the importance of tailoring surgical decisions to individual patient profiles, balancing clinical outcomes with guideline recommendations.

Consistent with the literature, the shift toward TL was associated with improved postoperative surgical outcomes [7,46,47]. Complication rates declined significantly over time, with global transient hypoparathyroidism rates dropping from 11% in 2014 to 3% in 2023. While the rate of RLN palsy did not differ significantly between groups, this finding is likely attributable to the low incidence of RLN injury, suggesting that this study may be underpowered to detect a significant difference. However, four cases of contralateral RLN palsy occurred in the TT group, underscoring the potential benefits of surgical de-escalation. These findings should nevertheless be interpreted with caution due to selection bias, as patients selected for TL generally presented with smaller, less aggressive tumors which likely contributed to the lower complication rates. Despite this, and in the absence of relevant changes in procedure-specific complication rates, the increasing adoption of TL may partly explain the observed reduction in the overall complication rate at the population level, largely driven by a lower incidence of postoperative hypoparathyroidism.

Given the lack of consensus on the best treatment for low-risk PTC, careful preoperative risk stratification is essential to identify patients suitable for TL. Individualized care, including informed discussions about the risks and benefits of TL versus TT, is increasingly recognized as critical for optimizing outcomes [11,13,48]. Although conducted at a single high-volume center, our findings align with previously reported international trends toward increased adoption of TL following the 2015 ATA guidelines [15,35,49].

Strengths of this study include its large sample size, coverage of a significant portion of the Italian population, and comprehensive evaluation of trends before and after guideline implementation. Several limitations should be acknowledged. The non-randomized allocation of surgical procedures represents an inherent limitation as the choice of surgical approach may have been influenced by concomitant pathologies and by patient preference—factors not evaluated in this study due to its retrospective design. Additionally, as a single-center study conducted within a high-volume tertiary referral institution, the generalizability of these findings to other surgical contexts may be limited. Moreover, long-term oncologic outcomes such as recurrence rates, disease-specific survival, and quality-of-life measures were not available. In addition, routine preoperative molecular profiling (e.g., BRAF or TERT promoter mutations) was not available for the majority of patients and therefore was not incorporated into surgical decision-making. Future prospective studies with longer follow-up and integrated molecular data are needed to further refine patient selection and confirm the long-term safety and quality-of-life outcomes of surgical de-escalation strategies.

## 5. Conclusions

The 2015 ATA guidelines have markedly influenced the management of PTC, promoting a more selective and individualized surgical approach in appropriately selected low-risk patients. In our center, we observed an increasing adoption of TL for low-risk PTC which was associated with a reduced overall postoperative complication burden at the population level. These findings describe changing patterns of care following guideline implementation and support a more individualized surgical approach for low-risk PTC.

## Figures and Tables

**Figure 1 curroncol-33-00026-f001:**
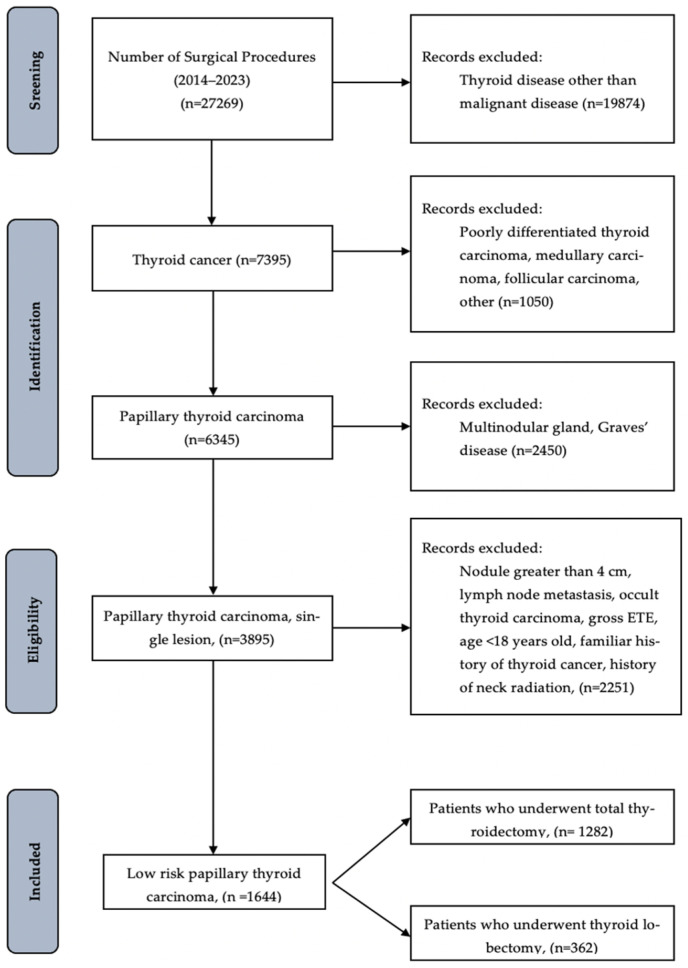
Study flowchart.

**Figure 2 curroncol-33-00026-f002:**
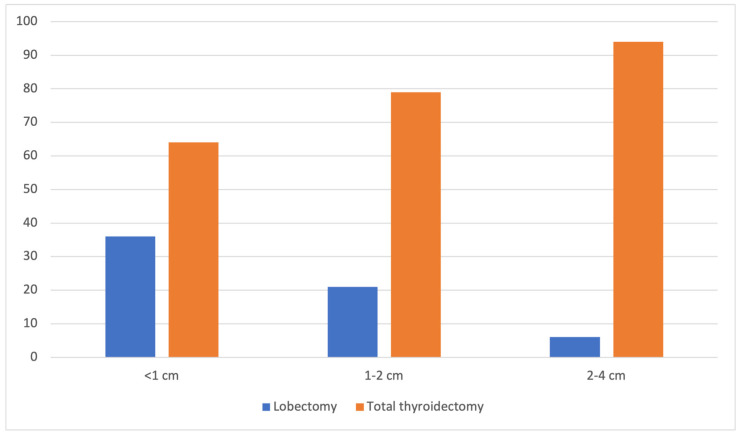
TT vs. TL rates by nodule size (TT—total thyroidectomy; TL—thyroid lobectomy).

**Figure 3 curroncol-33-00026-f003:**
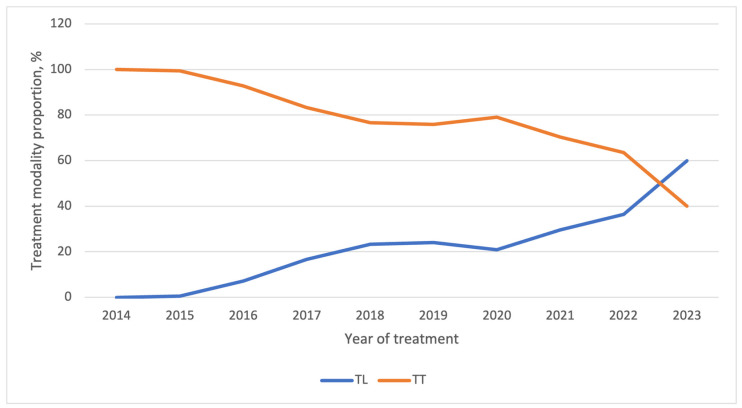
Global trends of TL and TT by year (TT—total thyroidectomy; TL—thyroid lobectomy).

**Figure 4 curroncol-33-00026-f004:**
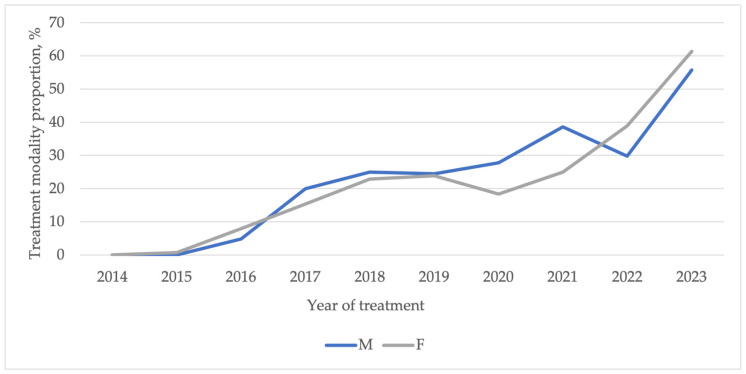
Temporal trends of thyroid lobectomy stratified by sex.

**Figure 5 curroncol-33-00026-f005:**
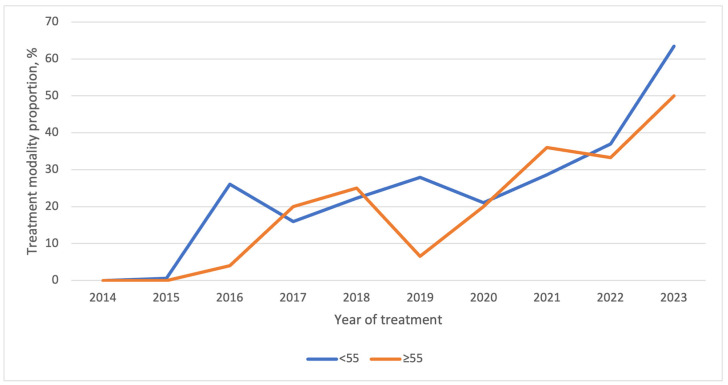
Temporal trends of thyroid lobectomy stratified by age at diagnosis.

**Figure 6 curroncol-33-00026-f006:**
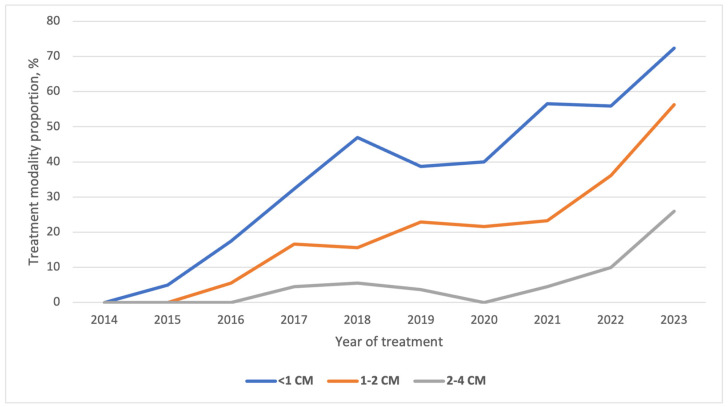
Temporal trends of lobectomy on the management of low-risk PTC stratified by nodule size.

**Figure 7 curroncol-33-00026-f007:**
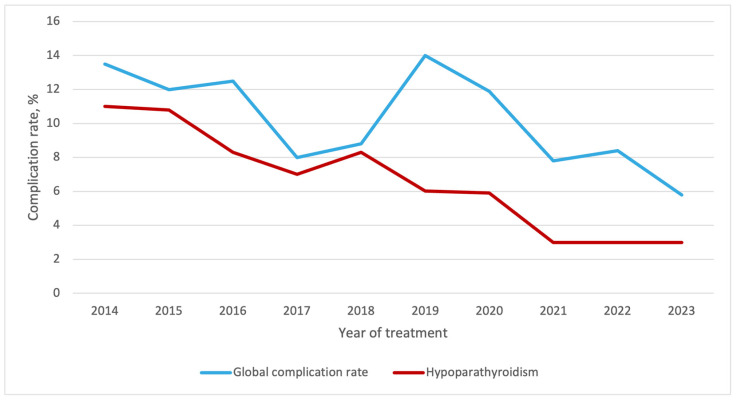
Temporal trends of postoperative outcomes of the entire population.

**Table 1 curroncol-33-00026-t001:** Clinical and demographic characteristics of the two groups (TL—thyroid lobectomy; TT—total thyroidectomy; ETV—Estimated thyroid volume; FNA—fine needle aspiration; BMI—body mass index; ns—not significant).

	Entire Population	TL	TT	*p*-Value
	1644	362	1282	
Sex				
M, (%)	426 (25.9)	101 (27.9)	325 (25.3)	0.32
F, (%)	1218 (74.1)	261 (72.1)	957 (74.7)	
Mean age in years (SD)	40.92 (13.19)	41.09 (13.4)	40.88 (13.14)	0.94
ETV in mL (SD)	13.67 (4.47)	11.88 (2.25)	14.32 (4.88)	**<0.001**
BMI (SD)	24.57 (4.45)	24.14 (4.08)	24.77 (4.60)	0.11
Preoperative FNAC (%)	1644 (100)	362 (100)	1282 (100)	ns
Preoperative cytology				
TIR4, N (%)	695 (42.3)	150 (41.4)	545 (42.5)	0.71
TIR5, N (%)	949 (57.7)	212 (58.6)	737 (57.5)	
Mean nodule dimension, mm (SD)	14.20 (7.01)	10.83 (4.3)	15.15 (7.33)	**<0.001**
Size, (%)				
<1 cm	443 (26.9)	159 (43.9)	284 (22.2)	**<0.001**
1–2 cm	879 (53.5)	185 (51.1)	694 (54.1)	
2–4 cm	322 (19.6)	18 (5)	304 (23.7)	
Age, (%)				
<55 anni	1388 (84.4)	301 (83.1)	1087 (84.8)	0.44
≥55 anni	256 (15.6)	61 (16.9)	195 (15.2)	

**Table 2 curroncol-33-00026-t002:** Comparison of postoperative outcomes between the two groups (TL—thyroid lobectomy; TT—total thyroidectomy; RLN—recurrent laryngeal nerve; MIVAT—minimally invasive video-assisted thyroidectomy).

	Entire Population	TL	TT	*p*-Value
Operative time, minutes (SD)	49.48 (17.1)	44.4 (20.1)	50.91 (15.9)	**<0.001**
Hospital stays, days (SD)	1.26 (0.5)	1.14 (0.4)	1.29 (0.6)	**<0.001**
Global complications, N (%)	173 (10.5)	11 (3)	159 (12.4)	**<0.001**
Transient Hypocalcemia, N (%)	115 (7)	0	115 (8.9)	**<0.001**
Permanent hypocalcemia, N (%)	23 (1.4)	0	23 (1.8)	**0.03**
Postoperative vocal cord dysfunction, N (%)	38 (2.3)	4 (1.1)	34 (2.7)	0.08
Contralateral RLN injury, N (%)	4 (0.2)	0	4 (0.3)	0.58
Hematoma, N (%)	19 (1.2)	7 (1.9)	12 (0.9)	0.11
**Approach**				
Conventional, N (%)	1414 (86)	261 (72)	1153 (89.9)	**<0.001**
Remote access, N (%)	92 (5.6)	50 (13.8)	42 (3.3)	
MIVAT, N (%)	138 (8.4)	51 (14.1)	87 (6.8)	
**Pathological Characteristics**				
Thyroiditis, N (%)	643 (39.1)	113 (31.2)	530 (41.3)	**<0.001**
Multifocality, N (%)	525 (31.9)	87 (24.1)	438 (34.1)	**<0.001**
Cancer on final histology, N (%)	1576 (95.8)	346 (95.6)	1230 (95.9)	0.87
Papillary, N (%)	1564 (95.1)	345 (95.3)	1219 (95.1)	0.97

**Table 3 curroncol-33-00026-t003:** Global trends on the management of low-risk PTC by year (TT—total thyroidectomy; TL—thyroid lobectomy).

YEAR	Number of Surgeries	TT, N (%)	TL, N (%)
2014	163	163 (100%)	0
2015	166	165 (99.4)	1 (0.6)
2016	167	155 (92.8)	12 (7.2)
2017	168	140 (83.3)	28 (16.7)
2018	180	138 (76.7)	42 (23.3)
2019	166	126 (75.9)	40 (24.1)
2020	134	106 (79.1)	28 (20.9)
2021	165	116 (70.3)	49 (29.7)
2022	165	105 (63.6)	60 (36.4)
2023	170	68 (40)	102 (60)

**Table 4 curroncol-33-00026-t004:** Completion rate based on nodule size.

	Lobectomies, N	Completion Rate, N (%)	*p*-Value
	362	50 (13.8)	
0–1 cm	159	22 (13.8)	0.34
1–2 cm	185	24 (13)	
2–4 cm	18	4 (22.2)	

## Data Availability

The data generated and analyzed during this study are available on reasonable request.

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
