# Peer review of "Evolving Practices in Low-Risk Papillary Thyroid Cancer: Impact of the 2015 ATA Guidelines"

_curroncol, 2026, doi:10.3390/curroncol33010026_

Round 1
Reviewer 1 Report
Comments and Suggestions for Authors
This is a well-designed retrospective cohort study examining the real-world adoption of the 2015 ATA guidelines regarding the surgical management of low-risk papillary thyroid carcinoma. The study is timely, clinically relevant, and methodologically sound. It provides valuable insights into trends toward less extensive surgery and associated outcomes over a 10-year period. The manuscript is generally well-written, the analysis is appropriate, and the conclusions are supported by the data. I recommend acceptance after minor revisions.
Comments to authors:
1. The study is retrospective and single-centered, which may limit generalizability. While the authors acknowledge this limitation, it would be helpful to contextualize how the findings from a high-volume Italian center compare with trends in other regions or healthcare systems.
2. The study focuses on surgical trends and short-term complications but does not include long-term oncologic outcomes such as recurrence rates, disease-specific survival, or quality of life measures. Given that the shift toward lobectomy is driven by equivalent long-term outcomes, the absence of such data is a notable gap. The authors appropriately mention this in the limitations, but future directions should emphasize the need for long-term follow-up studies.
3. The inclusion criteria are clearly described and align with ATA 2015 guidelines. However, there is no mention of whether molecular profiling (e.g., BRAF, TERT) was considered, which could further refine risk stratification. This could be noted as a limitation or future consideration.
4. The introduction provides a good background on thyroid cancer. However, it could be strengthened by including more recent references to highlight the current state of thyroid cancer classification and therapy (eg. Li et al, PMID:40057484).
Author Response
|
Comment 1: The study is retrospective and single-centered, which may limit generalizability. While the authors acknowledge this limitation, it would be helpful to contextualize how the findings from a high-volume Italian center compare with trends in other regions or healthcare systems. Response 1: We agree with the Reviewer that the single-center, retrospective design may limit the generalizability of our findings. To address this point, we have expanded the Discussion to contextualize our findings within international practice patterns and to emphasize that the observed trends are consistent with previously reported global shifts toward less extensive surgery following the 2015 ATA guidelines. (Discussion section, page 12, lines 336-338) Comment 2: The study focuses on surgical trends and short-term complications but does not include long-term oncologic outcomes such as recurrence rates, disease-specific survival, or quality of life measures. Given that the shift toward lobectomy is driven by equivalent long-term outcomes, the absence of such data is a notable gap. The authors appropriately mention this in the limitations, but future directions should emphasize the need for long-term follow-up studies. Response 2: We thank the Reviewer for this important observation. As suggested, we have reinforced the Limitations and Future Directions sections to emphasize the absence of long-term oncologic outcomes more explicitly, such as recurrence rates and disease-specific survival. We now clearly state that future studies with extended follow-up are warranted to confirm the long-term oncologic safety and quality-of-life outcomes associated with the increasing use of thyroid lobectomy. (Discussion section, page 13, lines 350-353) Comment 3: The inclusion criteria are clearly described and align with ATA 2015 guidelines. However, there is no mention of whether molecular profiling (e.g., BRAF, TERT) was considered, which could further refine risk stratification. This could be noted as a limitation or future consideration. Response 3: We appreciate this comment and agree that molecular profiling represents an important tool for refining risk stratification in papillary thyroid carcinoma. We have now acknowledged the absence of routine preoperative molecular testing in our cohort as an additional limitation, and we have included this aspect as a relevant consideration for future research aimed at further individualizing surgical decision-making. (Discussion section, page 13, lines 348-351) Comment 4: The introduction provides a good background on thyroid cancer. However, it could be strengthened by including more recent references to highlight the current state of thyroid cancer classification and therapy (eg. Li et al, PMID:40057484). Response 4: We thank the Reviewer for this helpful suggestion. We have updated the Introduction to include more recent references addressing contemporary thyroid cancer classification and management, including the suggested article (Li et al., PMID:40057484), to better reflect the current state of the field. (Introduction section, page 2, lines 73-77) |
Reviewer 2 Report
Comments and Suggestions for Authors
Dear Authors,
Your paper “Evolving Practices in Low-Risk Papillary Thyroid Cancer: Impact of the 2015 ATA Guidelines”, curroncol-4046552 is written well, in an understandable and readable manner. As long as studies evaluating the adoption of new ATA guidelines are valuable and desirable, there is one major issue with your paper: the primary grouping of patients (TL and TT groups) is not random, resulting in bias in findings.
What were the criteria for elucidation of the therapy approach? This is not clearly stated; please add it to the manuscript. I suppose that the doctors (clinicians, endocrine surgeons) decided to perform TL or TT depending on the patient’s status—meaning that low-risk patients with small, non-invasive PTC had TL. This, per se, influenced the results presented on page 4, lines 152 to 160, as well as in lines 166 to 172. Therefore, these lines should be written just as a description of the data, rather than actual findings.
Written as it is, only the frequency of TL application could be taken as an independent variable, so it could be commented on. Higher frequency of TL application with a lower rate of postoperative complications means that during the tested period, the selection of patients referred to TL was improved (as shown in figure 7 and table 4 and explained in lines 285-294 in the discussion section).
Consequently, the conclusion should be revised as the primary patient grouping contained selection bias—small and non-aggressive cases were selected for TL. As an expected outcome of this selection, patients in group A are less prone to the complications. This should also be addressed in the study's limitations section.
Long story short, please clearly define the criteria used by the doctors to determine whether a patient was referred to TL (belongs to group A) or TT (belongs to group B) and revise the relevant sections of the results, discussion, and conclusion.
Besides this major, here are the minor suggestions:
- Page 5, lines 193–196—Please, define the group of patients (TL or TT, or total sample?).
- Table 2—the unit of measurement (operative time, hospital stays) and the explanation of the MIVAT abbreviation are missing. Please add it.
- Table 3 – the symbol % is missing next to the column names. Please add it.
- Which group of patients is presented in figure 7? I suggest you show separately the trends in TL and TT groups and then compare the trends.
Author Response
|
Major Comment: Your paper “Evolving Practices in Low-Risk Papillary Thyroid Cancer: Impact of the 2015 ATA Guidelines”, curroncol-4046552 is written well, in an understandable and readable manner. As long as studies evaluating the adoption of new ATA guidelines are valuable and desirable, there is one major issue with your paper: the primary grouping of patients (TL and TT groups) is not random, resulting in bias in findings. What were the criteria for elucidation of the therapy approach? This is not clearly stated; please add it to the manuscript. I suppose that the doctors (clinicians, endocrine surgeons) decided to perform TL or TT depending on the patient’s status—meaning that low-risk patients with small, non-invasive PTC had TL. This, per se, influenced the results presented on page 4, lines 152 to 160, as well as in lines 166 to 172. Therefore, these lines should be written just as a description of the data, rather than actual findings. Written as it is, only the frequency of TL application could be taken as an independent variable, so it could be commented on. Higher frequency of TL application with a lower rate of postoperative complications means that during the tested period, the selection of patients referred to TL was improved (as shown in figure 7 and table 4 and explained in lines 285-294 in the discussion section). Consequently, the conclusion should be revised as the primary patient grouping contained selection bias—small and non-aggressive cases were selected for TL. As an expected outcome of this selection, patients in group A are less prone to the complications. This should also be addressed in the study's limitations section. Long story short, please clearly define the criteria used by the doctors to determine whether a patient was referred to TL (belongs to group A) or TT (belongs to group B) and revise the relevant sections of the results, discussion, and conclusion. Response: We thank the Reviewer for this important methodological observation. We agree that the allocation to thyroid lobectomy (TL) or total thyroidectomy (TT) was not randomized and reflected routine clinical practice. To address this concern, we have clarified that the choice of surgical approach was based on preoperative clinical evaluation and surgeon judgment in accordance with routine clinical practice. (Method section, page 3, lines 114-118) Importantly, we would like to emphasize that the primary objective of this study was to evaluate changes in real-world surgical practice patterns over time following the introduction of the 2015 ATA guidelines, and to describe the associated trends in postoperative outcomes. From this perspective, the non-randomized nature of treatment allocation represents an inherent characteristic of observational, practice-based research and does not undermine the main objective of the study. Accordingly, we have revised Results section (Results section, page 4, lines 168-173) and the Discussion section to explicitly acknowledge the presence of selection bias and to emphasize that differences in postoperative outcomes should be interpreted descriptively rather than causally. We further clarified that patients selected for TL generally presented with smaller and less aggressive tumors, which likely contributed to the lower observed complication rates. (Discussion section, page 12, lines 327-329) At the same time, we highlighted that the increasing adoption of TL over time may have contributed to a reduction in the overall complication burden at the population level, largely driven by a decrease in postoperative hypoparathyroidism, which represents the most frequent complication after TT. This interpretation aligns with the temporal trends observed in our data and does not imply a direct causal comparison between surgical techniques. (Discussion section, page 12 lines 329-332) Additionally, this issue has been explicitly addressed in the Limitations section. (Discussion section, page 12, lines 341-344) Relevant changes have been made in the Materials and Methods, Discussion, Limitations, and Conclusions sections. Minor comments Comment 1: Page 5, lines 193–196—Please, define the group of patients (TL or TT, or total sample?). Response 1: The text has been revised to clearly specify the patient group being referred to in this section (Page 5, line 204) Comment 2: Table 2—the unit of measurement (operative time, hospital stays) and the explanation of the MIVAT abbreviation are missing. Please add it. Response 2: Units of measurement for operative time and hospital stay have been added. The abbreviation MIVAT is now defined in the table legend. (Table 2, page 7, lines 219-220) Comment 3: Table 3 -The symbol % is missing next to the column names. Please add it. Response 3: The percentage symbol (%) has been added to the relevant column headings. (Table 3) Comment 4: Which group of patients is presented in figure 7? I suggest you show separately the trends in TL and TT groups and then compare the trends. Response 4: The figure legend has been revised to clearly indicate the patient population shown (Figure 7, page 10, line 242). As clarified in the Discussion, the observed reduction in overall complication rates reflects changes in the distribution of surgical procedures over time rather than variations in procedure-specific complication rates. For this reason, presenting postoperative outcome trends separately by surgical procedure was not considered to provide additional interpretative value. |
Round 2
Reviewer 2 Report
Comments and Suggestions for Authors
The manuscript has been sufficiently improved.